# On-admission SARS-CoV-2 RNAemia as a single potent predictive marker of critical condition development and mortality in COVID-19

Shoji Miki[1], Hiroaki Sasaki[2]*, Hiroshi Horiuchi[2], Nobuyuki Miyata[2], Yukihiro Yoshimura[2], Kazuhito Miyazaki[3], Takayuki Matsumura[4], Yoshimasa Takahashi[4], Tadaki Suzuki[5], Tetsuro Matano[1,6,7], Ai Kawana-Tachikawa[1,6,7], Natsuo Tachikawa[2]

1 AIDS Research Center, National Institute of Infectious Diseases, Tokyo, Japan, 2 Department of Infectious Diseases, Yokohama Municipal Citizen's Hospital, Yokohama, Kanagawa, Japan, 3 Department of Respiratory Medicine, Yokohama Municipal Citizen's Hospital, Yokohama, Kanagawa, Japan, 4 Research Center for Drug and Vaccine Development, National Institute of Infectious Diseases, Tokyo, Japan, 5 Department of Pathology, National Institute of Infectious Diseases, Tokyo, Japan, 6 Joint Research Center for Human Retrovirus Infection, Kumamoto University, Kumamoto, Japan, 7 Department of AIDS Vaccine Development, Institute of Medical Science, University of Tokyo, Tokyo, Japan

* sasaki.hiro.21@gmail.com

**Data Availability Statement:** All relevant data are within the paper and its Supporting Information files.

## Abstract

### Background

This study aimed to clarify how SARS-CoV-2 RNAemia is related to COVID-19 critical condition development and mortality in comparison with other predictive markers and scoring systems.

### Methods

This is a retrospective cohort study conducted at Yokohama Municipal Citizen's Hospital and National Institute of Infectious Diseases. We recruited adult patients with COVID-19 admitted between March 2020 and January 2021. We compared RNAemia with clinical status on admission including scoring systems such as the 4C Mortality, CURB-65, and A-DROP, as well as the $C_t$ value of the nasopharyngeal PCR, in predicting COVID-19 mortality and critical condition development.

### Results

Of the 92 recruited patients (median age, 58; interquartile range, 45–71 years), 14 (14.9%) had RNAemia. These patients had an older age (median, 68 years vs. 55.5 years; $p = 0.011$), higher values of lactated dehydrogenase (median, 381 U/L vs. 256.5 U/L, $p < 0.001$), C-reactive protein (median, 10.9 mg/dL vs. 3.8 mg/dL; $p < 0.001$), D-dimer (median, 2.07 μg/mL vs. 1.28 μg/mL; $p = 0.015$), lower values of lymphocyte (median, 802/μL vs. 1007/μL, $p = 0.025$) and $C_t$ of the nasopharyngeal PCR assay (median, 20.59 vs. 25.54; $p = 0.021$) than those without RNAemia. Univariate analysis showed RNAemia was associated

**Funding:** This work was supported by the Japan Agency for Medical Research and Development (AMED) under Grant Numbers JP19fk0108104j0801.

**Competing interests:** The authors have declared that no competing interests exist.

with mortality (odds ratio [OR], 18.75; 95% confidence interval [CI], 3.92–89.76; area under the receiver operating characteristic curve [AUC], 0.7851; $p$ = 0.002) and critical condition (OR, 72.00; 95% CI, 12.98–399.29; AUC, 0.8198; $p$ < 0.001). Plus, multivariate analysis also revealed the association of RNAemia with critical condition (adjusted OR, 125.71; 95% CI, 11.47–1377.32; $p$ < 0.001).

## Conclusion

On-admission SARS-CoV-2 RNAemia is a potent predictive marker of COVID-19 critical condition and mortality. The adjusted OR for critical condition was as high as 125.71.

## Introduction

COVID-19 pandemic is an ongoing global concern. Its clinical features range from an asymptomatic condition to critical diseases such as respiratory failure, shock, and multiple organ failure [1]. Predicting severity before deterioration is crucial for the timely treatment and appropriate management of patients, considering the limited medical resources during this overwhelming pandemic. Several risk factors, including lactated dehydrogenase (LDH), C-reactive protein (CRP), ferritin, and lymphocytopenia, have been proposed [2–4]. In addition, many risk scoring systems, such as the CURB-65 [5, 6], A-DROP [7, 8], and 4C Mortality [9], appear to be useful when predicting a fatal or critical condition but require reevaluation or validation because of the high risk of bias in the studies [10]. Furthermore, a clinically reliable prediction model remains urgently needed. In critical COVID-19 condition, systemic inflammation and multiple organ failure occur. In the analogy of various respiratory viral diseases, viral nucleic acids present in the blood is associated with a worse outcome; for instance, RNAemia of influenza A (H1N1) pdm09 or MERS-CoV infection is reportedly related to disease severity [11, 12]. In fact, RNAemia might play a role in the development of systemic inflammation in COVID-19 [13–16], but the clinical impact remains unclear. Hence, in this study, we analyzed the relationship between SARS-CoV-2 RNAemia and COVID-19 mortality and critical condition development. We also assessed whether RNAemia at the initial evaluation of hospital visit could be used as a predictor of a fatal or critical condition and to compare its utility with that of the preexisting prognostic indicators.

## Material and methods

### Study design and population

This retrospective cohort study was conducted at Yokohama Municipal Citizen's Hospital in Kanagawa, Japan, and National Institute of Infectious Diseases in Tokyo, Japan, and was approved by the ethics committee of these institutions. We recruited adult patients with COVID-19 aged ≥18 years who were admitted at the abovementioned hospital between March 2020 and January 2021. COVID-19 was verified by the positivity of the SARS-CoV-2 PCR assay of nasopharyngeal swab samples. Pregnant women were excluded. All eligible patients provided written informed consent for study participation.

### Detection of SARS-CoV-2 RNA in plasma and nasopharyngeal swab

Viral RNA was extracted from 140 μL of plasma and eluted with 30 μL of $H_2O$ using Viral RNA Extraction Kit (QIAGEN, Hilden, Germany). We utilized 5 μL of the extracted RNA for

the real-time RT-PCR of SARS-CoV-2, with the use of the N2 primer/probe set as previously described [17], in duplicate with QuantiTect Probe RT-PCR Kit (QIAGEN) and QuantStudio 5 real-time PCR system (ThermoFisher, Waltham, MA, USA). We repeated the real-time RT-PCR independently; thus, each sample was tested in four wells. If one of the wells demonstrated slope elevation during 45 cycles, the plasma sample was deemed SARS-CoV-2 RNA positive. The lower limit of the standard curve was 62.5 copies/reaction, and the $C_t$ value was almost 40. The SARS-CoV-2 RNA of the nasopharyngeal swab was detected by amplifying E gene with SARS-CoV-2 RT qPCR kit and Roche Light Cycler 480 (Roche, Basel, Switzerland) [18]. If the slope rose during 40 cycles, the sample was considered to be SARS-CoV-2 RNA positive.

## Definitions

The plasma samples were collected after confirming the informed consent to this study without delay from admission. Patients with a positive SARS-CoV-2 PCR assay in the plasma sample were categorized as the RNAemia group, whereas those with a negative assay were classified as the negative group. The negative group also included patients who were SARS-CoV-2 PCR negative in the first plasma but turned positive in the follow-up plasma.

The disease severity was defined according to the National Institutes of Health's *Coronavirus Disease 2019 (COVID-19) Treatment Guidelines* as follows [19]. Mild illness referred to the presence of signs and symptoms of COVID-19, excluding shortness of breath, dyspnea, or abnormal chest imaging. Having a lower respiratory involvement during clinical assessment or imaging and an oxygen saturation ($SpO_2$) of $\geq$94% on room air at sea level indicated moderate illness. For severe illness, the manifestation included $SpO_2 < 94\%$ on room air at sea level, a ratio of arterial partial pressure of oxygen to fraction of inspired oxygen ($PaO_2/FiO_2$) < 300 mm Hg, respiratory rate > 30 breaths/min, or lung infiltrates > 50%. Lastly, critical illness was defined as the occurrence of respiratory failure, septic shock, and/or multiple organ dysfunction.

## Data collection and statistical analysis

Using the electronic medical records system, we obtained patient data such as the demographic characteristics, the days since the onset, on-admission condition, and clinical data, such as the laboratory data including $C_t$ value of the PCR assay for SARS-CoV-2 E protein with the nasopharyngeal swab sample, disease severity during hospital stay, and in-hospital all-cause mortality. We also calculated the 4C Mortality, CURB-65, and A-DROP scores according to the on-admission condition. We then assessed the association of RNAemia with the on-admission condition and the outcome. All statistical data were analyzed using SAS Institute JMP version 15.2.1. Patients with missing data were excluded in each analysis. The $C_t$ value was considered to be 40 in the statistical analysis, if PCR assay result obtained from the nasopharyngeal sample on admission was negative: those patients were diagnosed as COVID-19 before the admission. The association of RNAemia with demographic characteristics, the days of sampling since the onset, and on-admission clinical data including scoring systems was evaluated using Mann-Whitney *U* test and two-tailed Fisher's exact test. We initially employed a univariate logistic regression model to evaluate the effect of each demographic characteristic and on-admission status on both critical condition and mortality. We then employed the multivariate logistic regression analysis for critical condition. Predictive variables were chosen by the forward stepwise method among significant variables in univariate logistic regression models, excluding 4C Mortality, CURB-65, and A-DROP scores because of strong correlation with other individual clinical indicators. Then, to assess the impact of remdesivir use on the

outcome of RNAemia group, we analyzed with two-tailed Fisher's exact test. In those analyses, $p < 0.05$ indicated statistical significance.

## Results

During study period, we forced to cease inclusion because of the shortage of human and facility resources between November 2020 and January 2021. As a result, of the 391 patients admitted, only 92 were eligible for the analysis (median age, 57.5 years; interquartile range, 45–71 years) (Fig 1). All the clinical raw data of the patients are presented in the S1 Table. This study population consisted of 59 males and 33 females, with 14 (15.2%) in the RNAemia group and 78 (84.8%) in the negative group (Table 1).

There was no significant difference in the days of plasma sampling from the onset of symptoms between two groups. Sex was also not significantly different between two groups. However, the RNAemia group was older than the negative group (median, 68 vs. 55.5 years, $p = 0.011$). Additionally, compared to the negative group, the RNAemia group had higher LDH ($p < 0.001$), CRP ($p < 0.001$), and D-dimer ($p = 0.015$) values but had a lower absolute lymphocyte count ($p = 0.025$). The $C_t$ value of the nasopharyngeal PCR was also lower in the RNAemia group than in the negative group ($p = 0.021$). Ferritin was not significantly different between two groups ($p = 0.109$). Furthermore, the 4C Mortality ($p < 0.001$), CURB-65 ($p = 0.001$), and A-DROP ($p < 0.001$) scores were higher in the RNAemia group than in the negative group.

Regarding the outcome, the RNAemia group had 12 critical cases including 6 deaths, 1 severe case, 1 moderate case, and no mild cases (Fig 2), whereas the negative group had 6

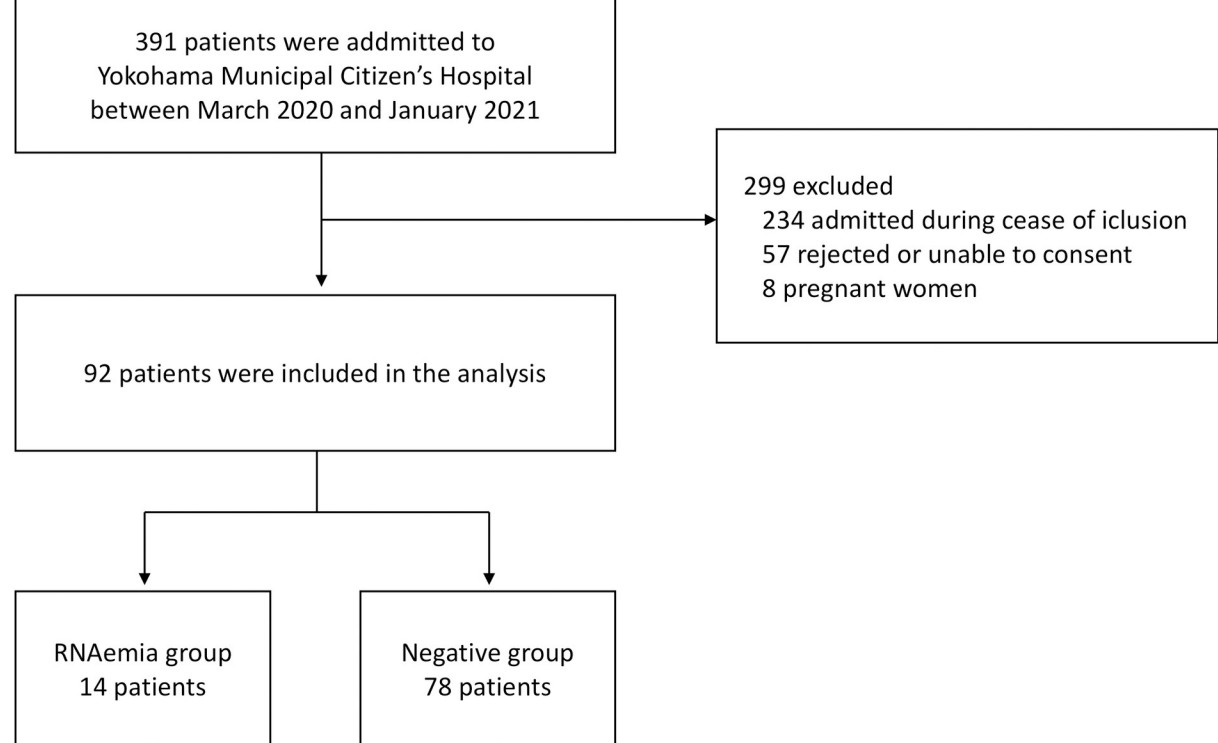

**Fig 1. Enrollment and result of plasma PCR.** The diagram represents the number of patients admitted to Yokohama Municipal Citizen's Hospital, patients enrolled into the study, and the result of plasma PCR assay.

**Table 1. Association between plasma PCR status and other clinical data.**

| | Plasma PCR | | |
| --- | --- | --- | --- |
| | **RNAemia (n = 14)** | **Negative (n = 78)** | ***p* value** |
| Days between symptoms onset and sampling | 8 [6.5–8.5] | 5 [2.25–8] | 0.052 |
| Demographic characteristics | | | |
| Sex (number of patients) | | | |
| Male | 9 | 50 | 1.000 |
| Female | 5 | 28 | |
| Age (years; Median [IQR]) | 68 [60–81] | 55.5 [44–70] | 0.011 |
| Laboratory data (Median [IQR]) | | | |
| LDH (U/L) | 381 [304.5–502] | 256.5 [200.25–327.5] | <0.001 |
| CRP (mg/dL) | 10.9 [8.1–16.4] | 3.8 [0.9–8.1] | <0.001 |
| D-dimer (μg/mL) | 2.07 [1.48–2.48] | 1.28 [0.97–1.96] | 0.015 |
| Ferritin (ng/mL) | 991.3 [216.3–2040.2] | 390.6 [147.0–881.0] | 0.109 |
| Absolute lymphocyte count (/μL) | 802 [521–1021] | 1007 [770–1389] | 0.025 |
| Ct value of the nasopharyngeal PCR | 20.59 [17.81–21.62] | 25.54 [20.94–32.2] | 0.021 |
| Risk scoring system (Median [IQR]) | | | |
| 4C Mortality Score | 11 [10–13] | 5 [3–9] | <0.001 |
| CURB-65 | 2 [1–3] | 0 [0–1] | 0.001 |
| A-DROP | 2 [1–3] | 0 [0–1] | <0.001 |

The *p* value was evaluated using two-tailed Fisher's exact test and Mann-Whitney *U* test. Abbreviations: IQR, interquartile range; LDH, lactate dehydrogenase; CRP, C-reactive protein.

critical cases including 3 deaths, 24 severe cases, 34 moderate cases, and 14 mild cases. 14 mild cases included two patients whose primary reasons of admission were other than COVID-19: One was drug eruption during chemotherapy for breast cancer and the other was stroke. In one critical but not fatal case in the negative group, plasma SARS-CoV-2 PCR turned positive 7 days after the admission. According to the outcome, RNAemia group accounted for 66.7% of fatal cases (6 out of 9), 66.7% of critical cases (12 out of 18), 4.0% of severe cases (1 out of 25), and 2.9% of moderate cases (1 out of 35). All the mild cases were found in the negative group.

By the univariate analysis, RNAemia group was associated with a higher mortality (odds ratio [OR], 18.75; 95% CI, 3.92–89.76; area under the receiver operating characteristic curve [AUC], 0.7851; $p < 0.001$) and critical condition development (OR, 72.00; 95% CI, 12.98–399.29; AUC, 0.8198; $p < 0.001$) (Table 2). Additionally, mortality was significantly associated with female sex (OR, 4.1481; 95% CI, 0.9633–17.8618; AUC, 0.6707; $p = 0.048$), older age (OR, 1.1835; 95% CI, 1.0695–1.3097; AUC, 0.9418; $p < 0.001$), high D-dimer level (OR, 1.5525; 95% CI, 1.0707–2.2509; AUC, 0.8889; $p < 0.001$), and low $C_t$ value of the nasopharyngeal PCR (OR, 0.6769; 95% CI, 0.5027–0.9114; AUC, 0.8656; $p < 0.001$). Meanwhile, critical condition was significantly associated with older age (OR, 1.0853; 95% CI, 1.0410–1.1315; AUC, 0.8247; $p < 0.001$), higher LDH level (OR, 1.0091; 95% CI, 1.0039–1.0143; AUC, 0.7508; $p < 0.001$), higher CRP level (OR, 1.1162; 95% CI, 1.0292–1.2107; AUC, 0.7455; $p = 0.007$), higher D-dimer level (OR, 1.3244; 95% CI, 0.9882–1.7748; AUC, 0.7734; $p = 0.004$), and lower $C_t$ value of the nasopharyngeal PCR (OR, 0.6698; 95% CI, 0.5080–0.8833, AUC, 0.8849; $p < 0.001$). In all the critical cases, the $C_t$ value of the nasopharyngeal PCR was <22. While all the patients with a $C_t$ value of <22 were critical in the viraemia group, only three out of 16 patients (18.8%) in the negative group were critical (Fig 3). Regarding the scoring systems, a higher rate was significantly associated with a higher mortality, with OR values of 1.7441 (95% CI, 1.2694–2.3964; AUC, 0.9532; $p < 0.001$), 4.5482 (95% CI, 2.0303–10.1889; AUC, 0.9244; $p < 0.001$),

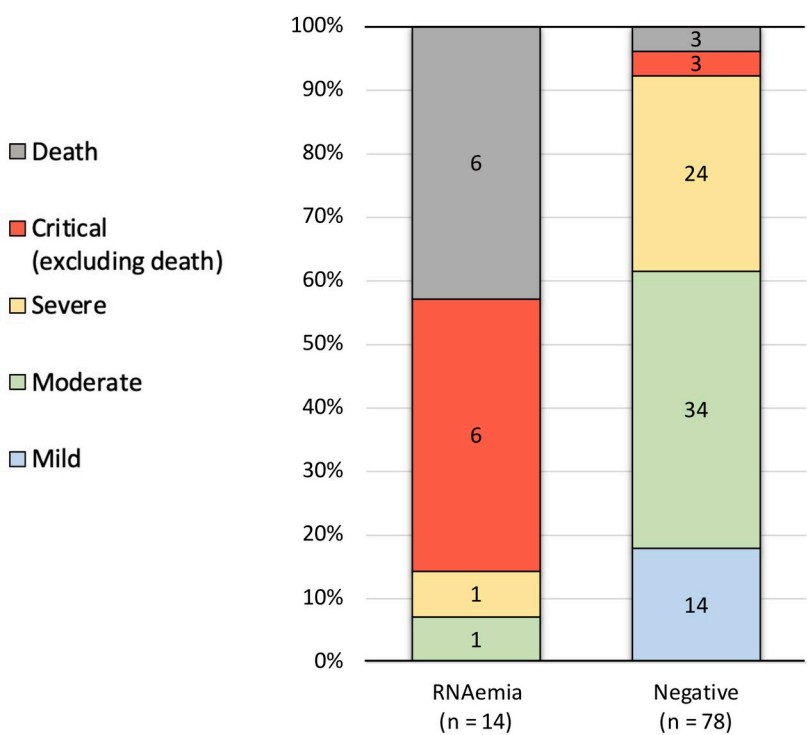

**Fig 2. Outcome of the patients in each group.** In the RNAemia group, 85.7% (twelve out of 14) were critical cases, including fatal cases, which accouted for 42.9% (six out of 14). In contrast, in the negative group, critical cases accouted for only 7.7% (six out of 78) and 3.8% (three out of 78) of the patients who died.

and 4.4331 (95% CI, 2.0729–9.4803; AUC, 0.9351; $p < 0.001$) for the 4C Mortality, CURB-65, and A-DROP scores, respectively. A higher score rate was also significantly associated with critical condition development, with OR values of 1.5507 (95% CI, 1.2601–1.9069; AUC, 0.8983; $p < 0.001$), 3.6584 (95% CI, 2.0649–6.4816; AUC, 0.8465; $p < 0.001$), and 3.7778 (95% CI, 2.1474–6.6462; AUC, 0.8904; $p < 0.001$) for the 4C Mortality, CURB-65, and A-DROP scores, respectively. By forward stepwise regression analysis, RNAemia, age, and D-dimer was chosen as an dependent variable for multivariate regression analysis for critical condition, which showed that RNAemia was most significant variable (adjusted OR, 125.71; 95% CI, 11.47–1377.32; $p < 0.001$) (Table 3).

Of the 14 patients in RNAemia group, two (50.0%) out of four patients who were treated with remdesivir died, while four (40.0%) out of ten who were treated without remdesivir died, but the mortality was not affected with the remdesivir use ($p = 1.000$).

Variables were chosen by the forward stepwise method among significant variables in univariate logistic regression models, excluding 4C Mortality, CURB-65, and A-DROP scores. Abbreviation: OR, odds ratio; CI, confidence interval.

## Discussion

This study demonstrated that SARS-CoV-2 RNAemia is a potent predictive marker of mortality and critical condition development in COVID-19. Although multiple clinical markers are significant by univariate logistic regression analysis, only three–RNAemia, age, and D-dimer–were left with the stepwise regression analysis and RNAemia was most significant. The adjusted OR for critical condition was as high as 125.71.

**Table 2. Univariate logistic regression analysis of each factor associated with mortality and critical condition.**

| On admission status | Mortality | | | Critical condition | | |
|---|---|---|---|---|---|---|
| | OR | AUC | *p* value | OR | AUC | *p* value |
| | (95% CI) | | | (95% CI) | | |
| Plasma PCR | 18.75 | 0.7851 | 0.002 | 72.0000 | 0.8198 | <0.001 |
| (positive to negative) | (3.9168–89.7584) | | | (12.9830–399.2905) | | |
| Sex (female to male) | 4.1481 | 0.6707 | 0.048 | 1.5680 | 0.5533 | 0.403 |
| | (0.9633–17.8618) | | | (0.5503–4.4677) | | |
| Age (years) | 1.1835 | 0.9412 | <0.001 | 1.0853 | 0.8247 | <0.001 |
| | (1.0695–1.3097) | | | (1.0410–1.1315) | | |
| LDH (U/L) | 1.0049 | 0.6633 | 0.098 | 1.0091 | 0.7508 | <0.001 |
| | (0.9992–1.0107) | | | (1.0039–1.0143) | | |
| CRP (mg/dL) | 1.0697 | 0.6968 | 0.196 | 1.1162 | 0.7455 | 0.007 |
| | (0.9686–1.1814) | | | (1.0292–1.2107) | | |
| D-dimer (µg/mL) | 1.5524 | 0.8889 | <0.001 | 1.3244 | 0.7734 | 0.004 |
| | (1.0707–2.2509) | | | (0.9882–1.7748) | | |
| Ferritin (ng/mL) | 1.0002 | 0.5890 | 0.664 | 1.0006 | 0.6343 | 0.036 |
| | (0.9994–1.0009) | | | (1.0000–1.0012) | | |
| Absolute lymphocyte count (/µL) | 0.9987 | 0.6827 | 0.108 | 0.9990 | 0.6201 | 0.094 |
| | (0.9970–1.0004) | | | (0.9978–1.0002) | | |
| Ct value of the nasopharyngeal PCR | 0.6767 | 0.8656 | <0.001 | 0.6698 | 0.8849 | <0.001 |
| | (0.5029–0.9106) | | | (0.5080–0.8833) | | |
| 4C Mortality Score | 1.7441 | 0.9532 | <0.001 | 1.5507 | 0.8983 | <0.001 |
| | (1.2694–2.3964) | | | (1.2610–1.9069) | | |
| CURB-65 | 4.5482 | 0.9244 | <0.001 | 3.6584 | 0.8465 | <0.001 |
| | (2.0303–10.1889) | | | (2.0649–6.4816) | | |
| A-DROP | 4.4331 | 0.9351 | <0.001 | 3.7778 | 0.8904 | <0.001 |
| | (2.0729–9.4803) | | | (2.1474–6.6462) | | |

Abbreviations: OR, odds ratio; CI, confidence interval; AUC, area under the receiver operating characteristic curve; LDH, lactate dehydrogenase; CRP, C-reactive protein.

COVID-19 symptoms range from an asymptomatic condition to severe pneumonia with respiratory failure, shock, and multiple organ failure [1]. As mentioned, predicting severity during the initial hospital visit is vital to promptly decide on the therapeutic intervention for each individual patient to prevent deterioration as well as on the appropriate medical resource management during this pandemic. To predict mortality risk, researchers have been using several scoring systems, such as the CURB-65, A-DROP, and 4C Mortality [6, 8, 9]. These scoring systems were useful in our study, consistent with previous studies. SARS-CoV-2 RNAemia, although single laboratory entity, was comparable with these scoring systems in predicting critical condition development in terms of high AUCs and ORs. Our study analyzed the $C_t$ value of the SARS-CoV-2 PCR from the nasopharyngeal swab sample because it is the gold-standard virological assay in diagnosing COVID-19. High viral load in the upper respiratory tract might cause RNAemia. Indeed, unlike previous study [16], the low $C_t$ value of the nasopharyngeal PCR was associated with RNAemia as well as fatal/critical condition. However, only few of our patients with relatively low nasopharyngeal $C_t$ values developed RNAemia and critical condition; thus, nasopharyngeal PCR could not replace plasma PCR in precisely recognizing RNAemia and predicting subsequent critical condition. In this study, we assessed RNAemia as an on-admission indicator of later disease development, and only RNAemia at

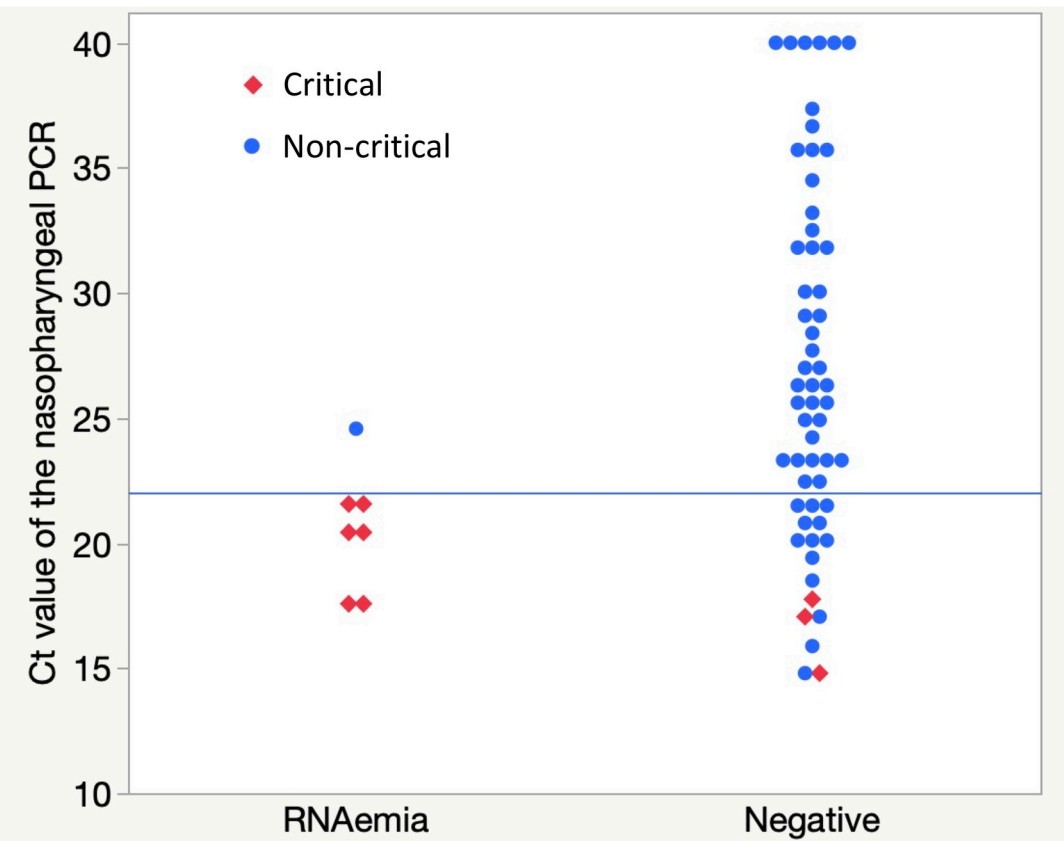

**Fig 3. Distribution of Ct value of the nasopharyngeal PCR in each group.** Each red rhombus represents critical cases including fatal cases, whereas each blue circle represents mild to severe cases. If nasopharyngeal PCR was negative, Ct value is represented as 40 in this figure. The median of Ct value of the nasopharyngeal PCR was significantly lower in RNAemia group than that in the negative group. While all the patients whose Ct value of nasopharyngeal PCR was <22 (represented as horizontal blue line) were critical in the RNAemia group, only three out of 16 those patients (18.8%) in negative group were critical.

the first plasma PCR was considered. In one critical case, however, plasma viral PCR turned positive after the initial plasma PCR assay, suggesting the importance of follow-up of plasma PCR during hospital stay. This issue remained to be dealt with, however, since we did not follow plasma PCR in all cases.

SARS-CoV-2 RNAemia and its association with the severity of COVID-19 has been reported in China, Spain, the United States, and Norway [13–16]. Consistent with previous reports, our study revealed that RNAemia is associated with a worse outcome among Japanese patients, possibly reflecting that this inclination is universal among various ethnicities. This report is the first to clarify the usefulness of RNAemia as a prognostic marker by directly comparing it with other clinical markers including the $C_t$ value of the nasopharyngeal PCR and the established scoring systems among the same patient group.

**Table 3. Mulivariate logistic regression analysis for the critical condition.**

| On admission status | adjusted OR (95% CI) | *p* value |
|---|---|---|
| Plasma PCR (positive to negative) | 125.71 (11.47–1377.32) | <0.001 |
| Age (years) | 1.1092 (1.0256–1.1997) | 0.001 |
| D-dimer (μg/mL) | 1.1137 (0.7855–1.5791) | 0.492 |

Our study has limitations. One major limitation is the small sample size, resulting in extremely wide confidence intervals of the odds ratios reported in the analysis, so that our quantitative result should be dealt with cautiously. The inadequate number of cases also restricted subgroup analysis. Our cases included various demographic and clinical characteristics, without considering important classifications such as specific age groups and immuno-compromised patients. Pregnant women were also excluded in our study. In addition, most of the included ethnicities were Japanese, so other ethnic groups were not considered; nevertheless, our results are consistent with previous reports focusing on other ethnicities, as previously described. Therefore, a large-scale multicenter, multinational study is needed to deal with these limitations.

Another limitation is the selection bias. We analyzed inpatients only; hence, many asymptomatic and mild cases were not evaluated. Furthermore, only 92 out of 391 inpatients were included in this study because of the cease of inclusion between November 2020 and January 2021, which marked the largest surge of COVID-19 cases in our community, due to the limited human and facility resources. Thus, certain selection bias could not be eliminated, although we had no intentions to include any specific condition.

Our study also lacks the adequate consideration of the treatment provided. RECOVERY trial is regarded as a breakthrough because it revealed the benefit of suppressing the inflammatory and immune reactions in severe and critical COVID-19 cases [20]. The cytotoxicity of SARS-CoV-2 is not so prominent compared with that of SARS-CoV-1 [21]; this finding explains the major cause of having a large population of asymptomatic patients, making the infection control of COVID-19 even more challenging worldwide. The pathophysiology of critical COVID-19 elicits an inappropriate response of the inflammatory and immune systems to the viral infection rather than the viral toxicity. Thus, immediate detection of the start of an inappropriate reaction is crucial. Based on the correlation of RNAemia and severity we herein revealed, we could pose a hypothesis that the invasion of a viral portion and/or the virus itself into the bloodstream could trigger such inappropriate inflammatory and immune reactions. If so, compared with other clinical indicators adopted in several scoring systems, RNAemia is the direct predictive marker of critical condition development. Antiviral treatments such as remdesivir and convalescent plasma have failed to demonstrate clear and major benefits in patients with COVID-19 overall but have appeared to be beneficial under certain conditions. For example, in ACTT-1 [22], which is a multinational, randomized, placebo-controlled trial, remdesivir reduced the time to recovery only in patients who were on low oxygen flow at baseline. Among critical patients who were on mechanical ventilation or extracorporeal membrane oxygenation at baseline, the time to recovery was similar to that of those with remdesivir therapy and placebo. Moreover, convalescent plasma showed no significant differences in clinical status or overall mortality between patients with COVID-19 overall [23]; however, when high-titer convalescent plasma against SARS-CoV-2 was early administered to mildly ill infected older adults, COVID-19 progression mitigated [24]. Given that such treatments are intended to reduce the virus itself rather than the host response, RNAemia might be the indicator necessary to make clinical decisions for initiating antiviral treatment. We failed to prove the superiority of remdesivir, probably because of too small number of cases with RNAemia. Further studies are warranted to verify our proposition.

## Conclusion

On-admission SARS-CoV-2 RNAemia is a potent predictive marker of mortality and critical condition development in hospitalized COVID-19 cases. The adjusted OR for critical condition was as high as 125.71.

## Supporting information

**S1 Table. All clinical data of the patients who were included in the present study.** Plasma PCR of patient No. 18 in the negative group turned positive on day seven. Abbreviation: No, number; CCI, Charlson Comorbidity Index; RR, respiratory rate; SBP, systolic blood pressure; DBP diastolic blood pressure; GCS, Glasgow Coma Scale; LDH, lactate dehydrogenase; BUN, blood urea nitrogen; CRP, C-reactive protein; ND, no data.
(XLSX)

## Acknowledgments

We thank Sachie Daigen for technical assistance.

## Author Contributions

**Conceptualization:** Shoji Miki, Yoshimasa Takahashi, Tadaki Suzuki, Tetsuro Matano, Ai Kawana-Tachikawa, Natsuo Tachikawa.

**Data curation:** Hiroaki Sasaki.

**Formal analysis:** Shoji Miki, Hiroaki Sasaki.

**Funding acquisition:** Ai Kawana-Tachikawa.

**Investigation:** Shoji Miki, Hiroaki Sasaki, Takayuki Matsumura.

**Methodology:** Takayuki Matsumura.

**Supervision:** Ai Kawana-Tachikawa, Natsuo Tachikawa.

**Writing – original draft:** Hiroaki Sasaki.

**Writing – review & editing:** Hiroaki Sasaki, Hiroshi Horiuchi, Nobuyuki Miyata, Yukihiro Yoshimura, Kazuhito Miyazaki, Takayuki Matsumura, Ai Kawana-Tachikawa, Natsuo Tachikawa.

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
