## [Decision Letter · Decision Letter 0]

14 May 2021

PONE-D-21-12753

On-admission SARS-CoV-2 viraemia as a single potent predictive marker of critical condition development and mortality in COVID-19

PLOS ONE

Dear Dr. Sasaki,

Thank you for submitting your manuscript to PLOS ONE. After careful consideration, we feel that it has merit but does not fully meet PLOS ONE’s publication criteria as it currently stands. Therefore, we invite you to submit a revised version of the manuscript that addresses the points raised during the review process.

We look forward to receiving your revised manuscript.

Kind regards,

Aleksandar R. Zivkovic

Academic Editor

PLOS ONE

Journal Requirements:

Reviewers' comments:

Reviewer #1: The authors present an observational cohort study of patients admitted to a hospital in Japan with COVID-19, relating biological and clinical parameters at admission to disease severity/mortality. The authors suggest that SARS-CoV-2 RT-PCR in plasma is a useful marker to predict a severe clinical course and in-hospital mortality, and potentially identify patients for antiviral therapy.

I have several comments:

1. As the authors state, the study is quite small, including 92 out of 391 COVID-19 inpatients during the inclusion period of 8 months. The authors do acknowledge the potential for selection bias – more information about why some patients were included and not others would be useful to evaluate this risk further. The manuscript cites human resources, while the flow chart cites a lack of consent. This should be clarified. With regards to systematic bias, do the authors have any data about patients not included in the study?

2. 14 out of 92 included patients had mild disease, and thus should normally not require hospitalisation. Was COVID-19 the cause of admission in all patients, or were some patients admitted for other diagnoses?

3. The authors give all data as median (25th/75th percentile), as for non-normal data, but the group comparisons are t-tests, which are generally used for normally-distributed data. Would Mann-Whitney tests be more suitable/robust?

4. There is no doubt that plasma SARS-CoV-2 RNA is strongly associated with severe disease and mortality in the presented data. However, evaluating the relative strengths of different predictors by directly comparing the odds ratios for mortality/critical illness doesn’t make sense – the OR for a binary classifier such as viremia will be much higher than for a continuous variable such as age or 4C Mortality score. Comparing the area under the ROC curve could be a better way of comparing the predictors with each other.

5. COVID-19 is a dynamic disease with viral and inflammatory phases. It would be useful to know the time from symptom onset to sampling.

6. The authors state that the first plasma sample after admission was analysed. Were these samples from the emergency department? First 24h of admission?

7. A number of patients have missing data for upper respiratory RNA measurements. Were they diagnosed with COVID-19 in another sample/before hospitalisation?

8. In the discussion the authors state that theirs is the first study to compare plasma RNA with upper respiratory CT value as a prognostic marker in SARS-CoV-2. This is inaccurate, Prebensen et al. published a study in Clinical Infectious Diseases in 2020, finding a strong prognostic value of plasma RNA but not nasopharyngeal RNA.

9. On lines 252-3 the authors seem to state that viral RNA in the circulation is the cause of inappropriate inflammation in COVID-19. Correlation is not causation, and I would suggest that the authors rephrase this, as the statement is not adequately supported by their data.

10. Finally, there are a number of typographical errors in the manuscript/tables, and I would recommend a thorough proof-reading before resubmission.

Reviewer #2: This is a useful study that supports other studies of a similar kind. For example, https://www.medrxiv.org/content/10.1101/2021.02.24.21252357v1

I have the following specific comments about the study:

1. The sample size is modest and incidence of viremia small. This is confirmed by the extremely wide confidence intervals of the odds ratios reported in the univariate logistic regression. This is a major limitation and makes this study primarily a hypothesis generating one rather than confirmatory.

2. The authors do not create a multivariate regression model (? cox proportional hazards or alternates). Careful selection of variables to create this model would be crucial to identify the additive value of measuring SARS CoV-2 plasma PCR in clinical practice.

3. What proportion of patients were initially negative by plasma PCR and later turned positive?

4. The lack of inclusion of therapeutic modalities is a major limitation as acknowledged by the authors? Is therapeutic data not available? It would be particularly interesting to see the impact of remdesivir on outcomes in the patients with viremia.

6. PLOS authors have the option to publish the peer review history of their article (what does this mean?). If published, this will include your full peer review and any attached files.

Reviewer #1: **Yes: **Christian Prebensen

Reviewer #2: No

---

## [Author Response · Author response to Decision Letter 0]

29 Jun 2021

Response to reviewers

Thank you for the thoughtful and constructive feedback you provided regarding our manuscript, PONE-D-21-12753, “On-admission SARS-CoV-2 viraemia as a single potent predictive marker of critical condition development and mortality in COVID-19.” 

We agree with your suggestions to revision of our manuscript and we have amended this to addresses the points from the reviewers, as well as based on our contemplation in order to better conform with the formatting and content rules of PLOS ONE.

With these changes to our final manuscript, we hereby resubmit our manuscript for a secondary evaluation. Thank you once again for your consideration of our paper. We would be glad to respond to any further questions and comments that you may have.

Change to the authorship

We added Takayuki Matsumura as an author, because of his contribution to our research by involving laboratory procedures.

Change to terminology

We changed the term “viraemia” to “RNAemia” in accordance with previous studies we cited in the manuscript. That’s because we have believed that the term “RNAemia” would be more precise, given that we did not isolate virus itself from blood i.e., viraemia but just showed the existence of RNA in blood.

Change to the reference list

We added reference No. 16, as mentioned below in the Response to Reviewer #1.

Responses to Reviewer #1

Thank you for providing for constructive insights.

Our responses to each comment are as follows.

1. As the authors state, the study is quite small, including 92 out of 391 COVID-19 inpatients during the inclusion period of 8 months. The authors do acknowledge the potential for selection bias – more information about why some patients were included and not others would be useful to evaluate this risk further. The manuscript cites human resources, while the flow chart cites a lack of consent. This should be clarified. With regards to systematic bias, do the authors have any data about patients not included in the study?

Response & Changes: Most of the excluded inpatients were hospitalized between November 2020 and January 2021, which marked the largest surge of COVID-19 patients in our community, that’s because we ceased inclusion to this study because of human and facility (mainly inadequate space of specimen storage) resources during the period. We clarified this pause of inclusion in the Figure 1 and manuscript lines 136–137, 271–273.

2. 14 out of 92 included patients had mild disease, and thus should normally not require hospitalisation. Was COVID-19 the cause of admission in all patients, or were some patients admitted for other diagnoses?

Response & Changes: The reason of admission of mild patients were miscellaneous. 2 out of 14 mild patients was admitted due to other diagnosis (drug eruption and stroke). We clarify this in the manuscript in lines 163–165. 6 mild cases were hospitalized because of relatively intense symptoms e.g., shortness of breath, general malaise, or decreased appetite, although categorized “mild” based on the NIH criteria. The others were admitted according to the government instructions at the given time for the purpose of quarantine.

3. The authors give all data as median (25th/75th percentile), as for non-normal data, but the group comparisons are t-tests, which are generally used for normally-distributed data. Would Mann-Whitney tests be more suitable/robust?

Response & Changes: We accepted the suggestion, reanalyzed, and revised the manuscript.

4. There is no doubt that plasma SARS-CoV-2 RNA is strongly associated with severe disease and mortality in the presented data. However, evaluating the relative strengths of different predictors by directly comparing the odds ratios for mortality/critical illness doesn’t make sense – the OR for a binary classifier such as viremia will be much higher than for a continuous variable such as age or 4C Mortality score. Comparing the area under the ROC curve could be a better way of comparing the predictors with each other.

Response & Changes: We added Area under the ROC curve as well as multivariate logistic regression analysis, to verify the superiority of plasma PCR to other predictors for the prediction of critical condition. However, AUC of the scoring systems such as 4C Mortality Score were higher than that of plasma PCR, so we changed the expression of superiority to comparable although it is single marker in lines 239–240.

5. COVID-19 is a dynamic disease with viral and inflammatory phases. It would be useful to know the time from symptom onset to sampling.

Response & Changes: We included the interval between onset and sampling, revealing there were no significant difference between two groups. We incorporated those findings into our manuscript in lines 152–153.

6. The authors state that the first plasma sample after admission was analysed. Were these samples from the emergency department? First 24h of admission?

Response & Changes: To be accurate, plasma samples used in this study were obtained after confirming the informed consent to this study. We revised the manuscript to clarify this in lines 98–99. As for most of the patients, the sample was obtained within two days. 

7. A number of patients have missing data for upper respiratory RNA measurements. Were they diagnosed with COVID-19 in another sample/before hospitalisation?

Response & Changes: Those patients were also diagnosed with the positive PCR status from nasopharyngeal swab sample, but it was with qualitative results and Ct values are not available. 

8. In the discussion the authors state that theirs is the first study to compare plasma RNA with upper respiratory CT value as a prognostic marker in SARS-CoV-2. This is inaccurate, Prebensen et al. published a study in Clinical Infectious Diseases in 2020, finding a strong prognostic value of plasma RNA but not nasopharyngeal RNA.

Response & Changes: Thank you for pointing out, we added this article as reference #16 and incorporated the comparison with previous study with regards to Ct value of the upper respiratory sample in line 244.

9. On lines 252-3 the authors seem to state that viral RNA in the circulation is the cause of inappropriate inflammation in COVID-19. Correlation is not causation, and I would suggest that the authors rephrase this, as the statement is not adequately supported by their data.

Response & Changes: We accepted the suggestion and clarified that the causation is nothing but a hypothesis in lines 238–284.

10. Finally, there are a number of typographical errors in the manuscript/tables, and I would recommend a thorough proof-reading before resubmission.

Response & Changes: Thank you for your recommendation.

Responses to Reviewer #2

Thank you for providing for constructive insights. As you pointed out, our study was consistent with several reports, and we cited several published articles in the manuscript.

1. The sample size is modest and incidence of viremia small. This is confirmed by the extremely wide confidence intervals of the odds ratios reported in the univariate logistic regression. This is a major limitation and makes this study primarily a hypothesis generating one rather than confirmatory.

Response & Changes: We accepted and incorporated this into the manuscript in line 260–262.

2. The authors do not create a multivariate regression model (? cox proportional hazards or alternates). Careful selection of variables to create this model would be crucial to identify the additive value of measuring SARS CoV-2 plasma PCR in clinical practice.

Response & Changes: We created the multivariate regression model adopting variables by the forward stepwise method among significant variables in univariate logistic regression models and showed statistically significant impact of plasma PCR and age on the outcome.

3. What proportion of patients were initially negative by plasma PCR and later turned positive?

Response & Changes: In one critical case, we recognized follow-up plasma PCR turned positive, but not all the patients’ plasma RNAemia were followed. We mentioned this in line 252.

4. The lack of inclusion of therapeutic modalities is a major limitation as acknowledged by the authors? Is therapeutic data not available? It would be particularly interesting to see the impact of remdesivir on outcomes in the patients with viremia.

Response & Changes: We evaluated the effect of remdesivir on mortality in RNAemia group with Fisher’s exact test, failed to reveal significant difference, apparently due to extremely small number of patients. We incorporated this into our manuscript.

---

## [Editor Report · Decision Letter 1]

1 Jul 2021

On-admission SARS-CoV-2 RNAemia as a single potent predictive marker of critical condition development and mortality in COVID-19

PONE-D-21-12753R1

Dear Dr. Sasaki,

We’re pleased to inform you that your manuscript has been judged scientifically suitable for publication and will be formally accepted for publication once it meets all outstanding technical requirements.

Kind regards,

Aleksandar R. Zivkovic

Academic Editor

PLOS ONE

---

## [Editor Report · Acceptance letter]

5 Jul 2021

PONE-D-21-12753R1 

On-admission SARS-CoV-2 RNAemia as a single potent predictive marker of critical condition development and mortality in COVID-19 

Dear Dr. Sasaki:

I'm pleased to inform you that your manuscript has been deemed suitable for publication in PLOS ONE. Congratulations! Your manuscript is now with our production department. 

Kind regards, 

on behalf of

Dr. Aleksandar R. Zivkovic 

Academic Editor

PLOS ONE